# The Responsible Health AI Readiness and Maturity Index (RHAMI): Applications for a Global Narrative Review of Leading AI Use Cases in Public Health Nutrition

**DOI:** 10.3390/nu18010038

**Published:** 2025-12-22

**Authors:** Dominique J. Monlezun, Gary Marshall, Lillian Omutoko, Patience Oduor, Donald Kokonya, John Rayel, Claudia Sotomayor, Oleg Sinyavskiy, Timothy Aksamit, Keir MacKay, David Grindem, Dhairya Jarsania, Tarek Souaid, Alberto Garcia, Colleen Gallagher, Cezar Iliescu, Sagar B. Dugani, Maria Ines Girault, María Elizabeth De Los Ríos Uriarte, Nandan Anavekar

**Affiliations:** 1Faculty of Bioethics, Universidad Anáhuac México, Mexico City 52786, Mexicoelizabeth.delosrios@anahuac.mx (M.E.D.L.R.U.); 2Division of Hospital Internal Medicine, Mayo Clinic, Rochester, MN 55905, USAmackay.keir@mayo.edu (K.M.); souaid.tarek@mayo.edu (T.S.);; 3Center for Responsible AI & Global Health, Global System Analytics & Structures, New Orleans, LA 70112, USA; 4Department of Educational Management, University of Nairobi, Nairobi 00200, Kenya; 5Africa Bioethics Network, Kigali 00502, Rwanda; 6School of Medicine, Masinde Muliro University of Science and Technology, Kakamega 190-50100, Kenya; 7College of Science, Bicol University, Legazpi City 4500, Philippines; 8Pellegrino Center for Clinical Bioethics, Georgetown University, Washington, DC 20007, USA; 9Department of Public Health, Asfendiyarov Kazakh National Medical University, Almaty 050012, Kazakhstan; 10Division of Pulmonary Medicine and Critical Care Medicine, Mayo Clinic, Rochester, MN 55905, USA; aksamit.timothy@mayo.edu; 11School of Bioethics, Ateneo Pontificio Regina Apostolorum, 00163 Rome, Italy; 12Honors College, University of Houston, Houston 77204, USA; 13Department of Cardiology, UT MD Anderson Cancer Center, Houston, TX 77030, USA; 14Department of Cardiology, Mayo Clinic, Rochester, MN 55905, USA

**Keywords:** public health nutrition, precision nutrition, artificial intelligence, ethics, responsible AI, swarm AI, edge AI, human rights, low- and middle-income countries

## Abstract

Poor diet is the leading preventable risk factor for death worldwide, associated with over 10 million premature deaths and USD 8 trillion related costs every year. Artificial intelligence or AI is rapidly emerging as the most historically disruptive, innovatively dynamic, rapidly scaled, cost-efficient, and economically productive technology (which is increasingly providing transformative countermeasures to these negative health trends, especially in low- and middle-income countries (LMICs) and underserved communities which bear the greatest burden from them). Yet widespread confusion persists among healthcare systems and policymakers on how to best identify, integrate, and evolve the safe, trusted, effective, affordable, and equitable AI solutions that are right for their communities, especially in public health nutrition. We therefore provide here the first known global, comprehensive, and actionable narrative review of the state of the art of AI-accelerated nutrition assessment and healthy eating for healthcare systems, generated by the first automated end-to-end empirical index for responsible health AI readiness and maturity: the Responsible Health AI readiness and Maturity Index (RHAMI). The index is built and the analysis and review conducted by a multi-national team spanning the Global North and South, consisting of front-line clinicians, ethicists, engineers, executives, administrators, public health practitioners, and policymakers. RHAMI analysis identified the top-performing healthcare systems and their nutrition AI, along with leading use cases including multimodal edge AI nutrition assessments as ambient intelligence, the strategic scaling of practical embedded precision nutrition platforms, and sovereign swarm agentic AI social networks for sustainable healthy diets. This index-based review is meant to facilitate standardized, continuous, automated, and real-time multi-disciplinary and multi-dimensional strategic planning, implementation, and optimization of AI capabilities and functionalities worldwide, aligned with healthcare systems’ strategic objectives, practical constraints, and local cultural values. The ultimate strategic objectives of the RHAMI’s application for AI-accelerated public health nutrition are to improve population health, financial efficiency, and societal equity through the global cooperation of the public and private sectors stretching across the Global North and South.

## 1. Introduction

Poor diet is the leading risk factor for death worldwide, associated with over 10 million premature deaths annually [1]. Excessive salt, sugar, and red and processed meats are linked to chronic inflammation, accelerating atherosclerosis, which underlies the expanding epidemic of non-communicable diseases (NCDs) of hypertension, diabetes, cancer, and cardiovascular disease. The United Nations Food and Agriculture Organization estimates that such unhealthy diets cost USD 8 trillion every year or nearly 10% of the world’s gross domestic product (GDP) [2]. Conversely, healthy diets followed by sufficient exercise, smoking avoidance, and moderated alcohol intake as part of the preventive social determinants of health (SDoH) drive approximately 80% of health outcomes, leaving 20% resulting from acute medical care treatments [3]. A total of 85% of clinicians report that they are not sufficiently trained or competent to provide healthy nutrition assessment, education, or support for their patients, with 75% of medical schools lacking any required nutrition classes for future clinicians [4]. Funding trends suggest that these health trends will only continue for the foreseeable future. Of the health sector’s worth of nearly USD 10 trillion, the United States accounts for nearly half of it by revenue and spending, which is almost twice as much per capita as other large developed countries’ share despite worse health outcomes—with almost 95% of American health expenditures going to healthcare systems delivering mostly acute medical care, with the rest going to public health that traditionally has specialized in disease prevention particularly through healthy diets [5,6]. Globally, 77% of the rising diet-related NCD deaths are concentrated in low- and middle-income countries (LMICs) in the Global South, with nearly half of them occurring in Africa, and this is where most of the shortage of 11 million healthcare workers is also concentrated [7,8]. While high-income countries (HICs) typically face the problem of an excess caloric intake contributing to unhealthy diets driving NCDs, LMICs face the double threat of this along with malnutrition from inadequate calories (impairing growth, development, cognition, and thus long-term overall health and economic productivity). Over 70% of Africa’s clinicians have emigrated to HICs in the Global North, mostly to the United States and the United Kingdom, and Africa already spends less on health than its debt service costs as it is trying to cross the Digital Divide to join the modern digitalized and industrialized global economy [9].

Solutions are underway. Artificial intelligence (AI) is emerging as history’s most disruptive, innovatively dynamic, rapidly scaled, cost-efficient, and economically productive technology that is increasingly providing transformative countermeasures to these negative health trends, especially in LMICs and underserved communities [10]. AI is revolutionizing industries globally. But adoption in the health sector continues to lag that in others, slowed by complex regulations, fragmented ecosystems, data security challenges, slow return on investment (ROI), and limited validated tools guiding implementation that can preserve patients’ trust and advances their wellbeing at affordable costs to them and their healthcare systems. We cannot improve what we cannot measure. Yet despite the proliferation of AI indexes for this, there are no validated full-spectrum ones to empirically inform the safe, effective, and affordable embrace of AI by the health sector, constituted by the healthcare, public health, and global health sub-sectors. Further, AI’s speed and complexity of development are largely driven by private AI platform companies that vastly outpace healthcare systems and even governments’ capabilities to define best practices and appropriately regulate them at the speed of relevance (amid shrinking public health and global health budgets, shunting more of the traditional societal needs of public health nutrition and wider prevention to them). Stanford University’s AI Index demonstrates the intense global competition to dominate the centralization of top AI: the United States owns nearly two-thirds of AI computing hubs, over 80% of advanced AI chips, nearly 70% of the leading models, and 12 times the private investment of the next competitor, China, which is making significant gains in catching up [11]. Governments pass laws, and institutions publish guidelines. But private platforms determine the bulk of daily AI priorities, operations, capabilities, applications, and trajectory. The World Health Organization (WHO) outlined its Global Initiative on AI for Health (GI-AI4H) with a particular LMIC focus to “harmonize governance standards” for AI as part of its larger strategic priority of accelerating the digitalization of healthcare systems worldwide through responsible, accessible, and sustainable AI in collaboration with aligned governments and companies [12]. Yet widespread confusion persists among healthcare systems’ clinicians, executives, and policymakers especially in LMICs on how to best identify, integrate, and evolve the safe, trusted, effective, affordable, and equitable AI solutions that are right for their communities. This is especially true in nutrition, which may be among the most effective fulcrums of public health advanced by healthcare systems, all the more so within LMICs and underserved communities which stand to benefit the most from such advances [10]. These dual research problems are exacerbated by the research gaps in the existing literature that lacks a rigorous global and multi-disciplinary review of leading AI use cases in public health nutrition, along with a practical tool for how to identity, integrate, and optimize them collaboratively within healthcare systems.

We therefore provide here the first known global, comprehensive, and actionable narrative review of the state of the art of AI-accelerated nutrition assessment and healthy eating, spanning public health nutrition for healthcare systems, generated by the first automated end-to-end empirical index for responsible health AI readiness and maturity. These twin innovations are meant to ultimately provide the practical contribution of assisting health actors to actually implement and benefit from the leading nutrition AI use cases (in a collaborative process across clinicians, executives, and policymakers within healthcare systems and partnering technology companies, public health agencies, governments, and non-government organizations in the Global North and Global South). This review seeks to do so through its academic contribution of filling the above research gaps while accelerating the translation of these results into health practice and workflows through the index. This review is contextualized in the larger evolution, digitalization, and platformization of the health ecosystem globally, using the index to map the likely trajectory of nutrition AI within the ecosystem through a unique full epistemic stack of scientific, economic, and ethical analyses. The index is built and analysis and review conducted by a multi-national team spanning the Global North and South, consisting of current and former front-line clinicians, ethicists, engineers, executives, administrators, public health practitioners, and policymakers. It builds on our team’s January 2025 policy analysis and recommendations solicited by the WHO for the responsible AI digitalization of LMIC healthcare systems [13]. This review codifies the earlier analysis and recommendations into a novel AI index which is then applied to a review of nutrition AI within healthcare systems for practically advancing the health of the populations they serve.

## 2. Responsible Health AI Readiness and Maturity Index (RHAMI) Design and Augmented Narrative Review

The first known full-spectrum composite AI index for health is described here—the Responsible Health AI readiness and Maturity Index (RHAMI)—to facilitate standardized, continuous, automated, and real-time multi-disciplinary and multi-dimensional strategic planning, implementation, and optimization of AI capabilities and functionalities worldwide, aligned with healthcare systems’ strategic objectives, practical constraints, and local cultural values. The ultimate strategic objectives of the RHAMI are to improve population health, financial efficiency, and societal equity through the global cooperation of the public and private sectors stretching across the Global North and South. The automated RHAMI can thus allow for longitudinal and customizable reviews of AI-augmented nutrition assessments and healthy eating advances even after this paper’s publication, along with other aspects of essential health services by healthcare systems for their communities. RHAMI automation is two-fold: hard (agentic AI orchestrator as a cloud-based API in Python 3.14.2 that can be embedded in existing dataflows and is modifiable for the needs of local healthcare systems with existing enterprise database management systems for real-time dashboard visualization, regardless of users’ technical expertise) and soft (education and calculator, including upskilling for staff, executives, and policymaker education on the RHAMI as an analytic framework in parallel with a customizable calculator with weights that can be modified according to local healthcare system needs).

The RHAMI was constructed by synthesizing the Stanford University Institute of Human-Centered Artificial Intelligence’s AI Index, the International Monetary Fund’s AI Readiness Index, the Oxford Government AI Readiness Index, Microsoft’s Health AI Maturity Roadmap, Access Partnership’s Responsible AI Readiness Index, IMD’s AI Maturity Index, the African Global Center on AI Governance’s Global Index on Responsible AI, and the GSAS AI-driven Efficiency-Inequity Index, which was then customized for the health sector [12,14,15,16,17,18]. Additional consideration was added to the typically overlooked system need of financial sustainability to improve effective planning and durable use enterprise-wide (or across the system as an organization), especially for LMIC healthcare systems and those integrated or partnered with public health and global health agencies. Continuous optimization testing was then conducted for the RHAMI to confirm that it exceeded the performance benchmarked to the above indexes in their respective domains from their related publications. The same iterative process was repeated with the RHAMI, which was then applied as a meta-index to confirm that streamlined end-to-end analyses accurately replicated primary consensus factors driving the top-performing healthcare systems globally (balancing the often competing objectives of the above individual indexes’ measures of patient access, health effectiveness, and financial sustainability within the ethical safeguards of responsibility to the common good).

In its final optimized formula, the RHAMI was engineered with two primary pillars: measuring the readiness of systems to deploy and scale AI and the maturity of its enterprise-wide depth and degree of advanced applications for this type of AI integrated with systems’ strategy, organizational structure, data infrastructure, culture, and governance (Figure 1). It spans four key domains: responsible AI (with sub-components of transparency, explainability, accountability, education, governance, and global bioethics), compliance (with sub-components of ethics, law, regulation, guidelines, and policy), organization (with sub-components of enterprise penetration, workflow integration, executive engagement, workforce development, strategic centrality, and digital infrastructure), and sustainability (with sub-components of ROI, operational risks, health outcomes, administrative outcomes, and ecological impact). The index functioning as a quantitative assessment was normalized to 100 points with 50 specific criteria in total. Given the various and often ambiguous definitions of ‘responsible AI’, the RHAMI featured the first end-to-end epistemic model in this topic that progressively builds from Personalist Quantum Metaphysics to Personalist Social Contract ethics to Personalist Liberalism (moral political economics) to Personalist AI Governance, providing a novel analytic incorporation into this empirical index of global political economic, multi-sector, and multicultural drivers of AI, nutrition, and health [17]. It operationalizes the WHO’s One Health human security-based approach by integrating the modern rights-based social contract ethics embodied in the only worldwide institutionalized moral consensus (codified by the United Nations’ 1948 Universal Declaration of Human Rights from which the WHO’s Health Organization, international law, and all major AI governance frameworks derive), anchored in classical Aristotelian realist metaphysics (with Thomistic personalist refinement) to uniquely allow for the substantive and practical convergence of the world’s diverse belief systems underlying our political economics as the meta-determinants of health, ultimately by safeguarding individual dignity fulfilled in commitment to the common good of the global human family, which in turns safeguards the good of each individual. The RHAMI accordingly maps how the form and function of the health ecosystem globally are a network of public–private partnerships driving AI infrastructure provisioned by user-facing platforms, expanding healthcare systems’ technical scale, clinical effectiveness, cost efficiency, and societal equity of their collective efforts generating AI-accelerated public health nutrition. Finally, the index was optimized according to the AI technique of reinforcement learning from human feedback and then operationalized for healthcare systems scaled to global health according to the template by the Boston University’s WHO-backed pioneering BEACON platform as a free, real-time, open-source, decentralized global biothreat surveillance platform with expert human-in-the-loop quality control [19]. AI-unassisted literature review and AI-assisted literature review were compared in terms of the root mean squared error (RMSE), as was RHAMI performance benchmarked based on the performance of the above indexes given the absence of a widely recognized gold standard for or published demonstration of such an integrated index.

An AI-unassisted comprehensive literature review was then performed from 2005 to 2025 using EBSCO, Web of Science, Scopus, Google Scholar, PubMed, and the citations of identified studies by experts specializing in the subject matter. Article selection based on quality and impact was aided by the PRISMA (Preferred Reporting Items for Systematic Reviews and Meta-Analyses) reporting standard. To better democratize and sustain this methodology for more healthcare system actors worldwide regardless of technical background, we also performed an AI-assisted literature review using the free ChatGPT 4o large language model (LLM), in addition to reviewing news articles, social media, books, and Common Crawl for the most up-to-date advances. In both review tracks, the search focused on search terms associated with nutrition and AI, including the following: “nutrition”, “nutrition assessment”, “dietary assessment”, “healthy eating”, “healthy nutrition”, “healthy diet”, “clinical nutrition”, “personalized nutrition”, “precision nutrition”, “nutrition education”, “diet education”, “artificial intelligence”, “machine learning”, “deep learning”, “reinforcement learning”, “computer vision”, “agentic AI”, “orchestrator AI”, “spatial AI”, artificial general intelligence”, and “AGI”. To refine the search, AND/OR as Boolean operators were used. Hard RHAMI automated analysis was then deployed on the AI-unassisted and -assisted literature review tracks to identify and stratify leading responsible AI use cases in nutrition according to readiness and maturity within and with healthcare systems.

## 3. RHAMI Identification and Stratification of Leading Healthcare Systems and Their Nutrition AI

The highest readiness and maturity across the greatest concentration of healthcare systems for adopting responsible AI for nutrition were in the United States, followed by Europe and the Asian countries of Japan, South Korea, and India, along with the United Arab Emirates (with which American healthcare systems and technology companies have historically dense institutional partnerships and supply chains across healthcare and technology). Within the health sector, most healthcare systems are at the earliest stages of AI maturity including being aware of or experimenting with AI but not yet operationalizing, scaling, nor transforming their systems with it. Within healthcare systems, Mayo Clinic, as the world’s largest integrated non-profit medical group, has the most mature enterprise-wide responsible health AI, after launching the first comprehensive AI-augmented Mayo Clinic Platform in support of its care network stretching across 60 healthcare organizations globally. Among AI platforms in parallel and partnering with healthcare systems, the top-performing ones for nutrition included America’s SnapCalorie and MyFitnessPal, the United Kingdom’s Zoe, Canada’s Nutrigenomix, India’s HealthifyMe, and South Korea’s Samsung Food. Healthcare systems principally partnered with individuals and populations using such nutrition AI platforms to fill technical capability gaps in their nutrition assessments and healthy eating support. But there were no healthcare systems with mature enterprise-wide nutrition AI, either in standardized and scaled best practices in nutrition assessments or in healthy eating education and support. Overall, the primary application areas for leading systems’ public health nutrition AI included the following: edge AI nutrition assessments, scaled precision nutrition, and sustainable healthy diets detailed below.

### Multimodal Edge AI Nutrition Assessments as Ambient Intelligence

Nutritional assessments are the historical cornerstone for informing individual behaviors and population policies facilitating healthier eating, by first detailing the amount, types, and timing of food intake (typically in the context of individuals’ height, weight, body mass index, medical conditions, physical exam, and laboratory tests to identify nutrition imbalances, excesses, and deficits) [20]. Yet they are notoriously difficult to accurately, quickly, consistently, and affordably conduct at a population scale. If one does not know where the nutrition problems are, then they do not know how to personalize the individual interventions and population policies to address them. Healthcare systems worldwide, especially in LMICs, lack the necessary number of dieticians skilled in these assessments, and healthcare providers generally lack sufficient and standardized education and practice with them. This is where AI is rapidly gaining momentum. Early use cases focused on labor and cost-intensive hand-built machine learning algorithms to replicate the traditional step-wise nutrition assessment [21]. By the late 2010s, deep learning algorithms became more widespread as more complex models automatically learned more relevant features faster and directly from raw data with less human programming, mostly achieving adequate performance in energy estimation and macronutrient tracking but lagging in micronutrients. More advanced recent deep learning deploys computer vision such as on the digital applications or apps of smartphones, functioning as edge digital devices linked to their cameras to provide users real-time assessments [22]. Wearable smart watches, glasses, and e-buttons provide even more complex ambient monitoring through motion sensing. The current leading use cases are coalescing around such edge AI nutrition assessment as ambient intelligence. This trend brings together edge computing (more private, secure, and energy-efficient local data processing and AI applications such as smartphone apps to reduce latency, bandwidth use, and cloud reliance at the edge of the global digital ecosystem), AI-enabled nutrition assessments (computer vision, audio detection, motion sensors, voice logging, smart food appliances, and wearable sensors to identify, quantify, and categorize foods consumed and their related nutrients and eating behaviors in progressively more automated and augmented fashion), and ambient intelligence (running such AI in the background without regular human commands). A 2024 analysis of EgoDiet (based on a mask region-based convolutional neural network) demonstrated that passive monitoring with low-cost AI-enabled wearable cameras was superior to traditional self-reported and dietician-conducted nutrition assessments for healthy portion sizes of local African cuisines in rural Ghana [23]. This culturally sensitive ambient intelligence enables more precise and personalized nutrition assessments and thus recommendations for healthy eating based on tracked individual food preferences, portion sizes, and meal timing. A 2025 feasibility clustered randomized controlled trial in the US demonstrated that primary care physicians utilized a multi-lingual AI expert clinical decision support (CDS) for nutrition assessment on 100% of outpatient encounters in lower-resourced safety net clinics, up from the historic threshold of 30–40% for expert CDSs required to be considered successful enough to scale for widespread clinical use [24]. This system, Nutri, was co-designed with clinicians, patients, and AI engineers to provide automated multi-level behavioral nutrition assessments and recommendations that can reliably be incorporated into brief but effective counseling segments of less than 3 min within a typical provider–patient visit. Nutri rapidly assesses and personalizes recommendations from the latest guidelines and research that clinicians can then modify and apply for achievable diet modifications while complementing rather than disrupting the other core aspects of clinical visits. A 2025 study in China showed how an LMIC hospital successfully deployed an extreme gradient boosting (XGBoost 3.1.1) AI algorithm using the GLIM (Global Leadership Initiative on Malnutrition) framework running in the background on EHRs for successful malnutrition screening, with the top six predictors identified being decreased food intake, weight loss, BMI, white cell count, neutrophil percentage, and pre-albumin levels [25]. A 2024 randomized controlled trial across 11 hospitals and 5763 patients in China showed that an AI-based rapid nutritional diagnostic system embedded within routine clinical care significantly improved malnutrition diagnosis and reversal, at an AI intervention cost that is 73% cheaper than China’s per capita cost of malnutrition’s health effects [26].

These studies provide notable evidence of practical sustainability, workflow embeddedness, clinical effectiveness, and cost efficiency. Yet none of these use cases nor any AI-enabled nutrition assessments are in standard or sustainable enterprise-wide deployment within healthcare systems as part of comprehensive nutrition care. And none outside of Nutri provide substantive detail behind their governance safeguards to assess let alone ensure responsible AI deployment. In RHAMI analysis, a feasible and affordable integrated nutrition assessment would marry use cases such as the American Nutri-like outpatient expert system with the Chinese inpatient malnutrition screen, paired with a smartphone or smartglasses version of the African EgoDiet for longitudinal and personalized ambient nutrition assessments (with embedded compliance-by-design governance as with Nutri based on the US HIPAA, or Health Insurance Portability and Accountability Act, the world’s leading regulatory framework for protecting individual health data, along with the mostly comparable European GDPR, or the General Data Protection Regulation). But most healthcare systems in the Global South significantly lag their northern counterparts in EHR adoption and functionality. Less than 25% of African LMIC healthcare systems have EHRs [27], but already over 65% of Africans have internet-capable smartphones, and this is on course to reach over 90% by 2030 [28]. AI platforms may thus extend the limited reach of healthcare systems as with EgoDiet or similar tools like MyFitnessPal (summer release 2025) [29]. This serves as the top nutrition app on the digital edge with over 250 million users in 120 countries, generating personalized nutrition plans from over 5 million foods based on this free tracker (for calories, macronutrients, and micronutrients enabled with computer vision, along with the fee-based subscription including barcode scanning and voice logging also).

## 4. Strategic Scaling of Practical Embedded Precision Nutrition Platforms

The results of AI-accelerated nutrition assessments are meant to inform actionable recommendations, with more personalized ones being more likely to ultimately succeed in improving health. At the most advanced end of that spectrum, precision nutrition arose out of translational multi-omics in precision medicine and precision public health to individualize dietary recommendations, including according to health status, lifestyle, microbiome, and genetics [30]. This trend is fueled by the historic successes, in particular, of precision medicine for cancer care to translate granular knowledge of individuals’ genes, lifestyles, and environments into targeted treatments with the maximum clinical benefit, minimum risk, and optimized overall affordability (uniting genomics, epigenomics, proteomics, metabolomics, and other omics biological fields to facilitate healthier states faster, thus reducing often unaffordable longitudinal clinical and financial costs of disease) [17]. But despite promising individual results, precision nutrition has been slow to scale even in HICs amid the substantive upfront investment, funding, infrastructure, and training required, thus limiting its population impact especially among LMICs. There are several studies advancing isolated but effective use cases of AI-enabled precision nutrition to accelerate its scaling throughout healthcare systems. A 2024 study in Greece not only demonstrated that a novel LLM with deep generative AI could accurately produce guideline-based personalized dietary recommendations over 84% of the time (among 1000 real profiles and 7000 daily meal plans), but it also showed that it could be paired with the more globally accessible ChatGPT at over 700 million weekly active users to improve accuracy and meal variety across more local cuisines worldwide [31]. The novel model accounted for individuals’ medical conditions, anthropometric measurements, and caloric needs with consensus recommendations as by the WHO. In 2023, a US team showed that ChatGPT could effectively generate meal plans customized for pregnant women from underserved populations [32]. Considered together, the Greek study may support how AI-enabled personalized nutrition through free open-source leading frontier AI platforms can produce comparable results to custom-built proprietary models for particular benefit to LMIC healthcare systems while also suggesting the feasibility for multi-omics data to be added as another data input like the above health data to move from personalized to precision nutrition at scale. A 2025 Stanford University analysis provided a more mature AI-augmented precision nutrition intervention integrated within its healthcare system for newborn children admitted to neonatal intensive care units requiring total peripheral nutrition (TPN) [33]. TPN can involve life-saving interventions infused through blood vessels to provide hospitalized patients nutrition when they cannot receive it through their digestive tract. Yet ordering TPN requires specialized knowledge with dieticians, pharmacists, and physicians collaborating on building these complex prescriptions, thus prone to error and complications. The team built TPN2.0 as a deep learning approach using a variational neural network with semi-supervised iterative clustering on an EHR dataset of routinely collected variables (including diagnoses, demographics, and TPN details across 79,790 TPN orders from 5913 patients, followed by model validation with a second hospital’s dataset). TPN2.0 identified 15 customized TPN formulas according to the unique needs of newborns grouped precisely into similar clusters. A blinded analysis within the trial showed that physicians were over 3 times as likely to rate TPN2.0 orders as superior to non-AI TPN orders crafted by traditional best practices. It also showed the AI orders had notably less risk of complications (including sepsis and death). The authors explicitly investigated the potential of scaling TPN2.0 globally to LMICs, as the clinical adoption of AI in neonatology is non-existent, and customized TPN is generally inaccessible at the population scale for LMICs. By balancing clustered treatment standardization with clinically relevant personalization, the Stanford model could “enable a precision-medicine approach” for nutrition that is safer to deploy, easier to order, and cheaper to generate (through mass manufacturing of TPN formulas and embedded use of TPN2.0 within healthcare systems’ existing EHRs).

A 2025 review by a US and Bangladesh team detailed the latest applications of precision nutrition in low-resource settings with a particular focus on such maternal and child health, citing the above Stanford study [34]. Healthcare systems especially in LMICs generally focus on public health nutrition by identifying malnutrition, supplementing micronutrients, and fortifying foods to prevent and reverse the undernutrition common in Stanford’s critically sick newborns and even more so in low-resource communities worldwide. Such AI uses of public health precision nutrition enable more impactful resource prioritization not only in intensive care units but more generally across populations by focusing on diseases that can benefit from more targeted and affordable nutrition interventions. The US and Bangladesh team additionally highlighted how AI-enabled microbiota-directed complementary foods produced twice the mean rate of growth compared to traditional ready-to-use therapeutic foods for severe acute malnutrition (though the study lacked demonstration of longitudinal follow-up after 3 months, cost efficiency, and healthcare system scale-up). But the LMIC analysis also pointed to the gradual introduction into the precision nutrition of AI-powered personalized digital twins to virtually simulate various customized nutrition interventions for patients according to their specific health status, microbiome, biochemistry, social settings, and environment. This approach can ultimately empower healthcare systems working in parallel (and preferably in coordination of limited health and digital resources) to better improve public health nutrition through precision nutrition scaled up to the population level. Doing so across HICs is a struggle, and among LMICs, it is nearly absent. But translational research for more equitable precision nutrition is expanding. The US National Institutes of Health (NIH) has unveiled within its All of Us program the largest AI-augmented precision nutrition research project across 14 healthcare systems and universities with 10,000 individuals, particularly drawn from historically underrepresented and underserved groups [35]. Over 53 million Americans are immigrants, as the United States is home to nearly 20% of all the world’s migrants (and history’s highest and longest immigration flow). The NIH’s overarching All of Us was therefore launched early in 2018 and has since built the largest and most diverse health dataset and precision health study ever, monitoring multi-omics data longitudinally on over 1 million people (with 80% identifying with racial and ethnic minorities who historically have been underrepresented in health research, and thus derivative drugs and procedures can be studied). The US government behind the NIH provides nearly half of all global health funding (nearly 4 times more than that of the next donor country, which is the United Kingdom) [36,37]. Most of that USD 12.4 billion annually is concentrated in the Global South, with 84% going to Africa alone, though only 1% or USD 168 million goes to nutrition (among the smallest funding priorities). But at nearly 4 times the global health budget, the US contributes USD 50 billion annually into biomedical research as the world’s largest funder of it—8 times the collective amount of the next five donors, which are British, Canadian, Australian, and European institutions and governments [38]. But worsening debt crises, distress, and pressure worldwide mean that more efficient prioritization on more impactful translational research is required, especially in nutrition, which holds the greatest return on investment for public health. In RHAMI analysis, such institutional funding structures are central nodes in the public–private networks constituting the health ecosystem, having a disproportionate influence on setting the research agenda, funding priorities, and practical health applications. The AI-enabled precision nutrition within All of Us therefore potentially represents a potent template and impetus for expanding inclusivity, relevance, and impact globally especially in LMICs for such applications (with an emphasis on those that can scale throughout healthcare systems even after study periods and their funding cycles conclude). But for now, RHAMI analysis also highlights that there are still no mature or even system-wide AI-enabled precision nutrition capabilities in HICs and no clear enduring use cases in LMICs. The study at Stanford (which also houses the Stanford Institute of Human-Centered AI) is limited to the United States with their financially and resource-intensive neonatal intensive care units. But it does represent the most clinically integrated, financially sustainable, and advanced governance demonstration of precision nutrition. It also suggests an underexplored trajectory gaining momentum—the strategically sustainable scaling of patient-first rather than AI-first practical precision nutrition. The goal of AI-enabled precision nutrition is not AI for AI’s sake, or precision nutrition for its own sake, but the ultimate public health achieved by the effective coordination and alignment of healthcare systems, empowered by AI improving the care they provide more widely, affordably, and sustainably. TPN2.0 tries to find and fix a current deficit in services by healthcare systems such as safe, affordable, and scalable TPN. It reverses these deficiencies with AI-enabled solutions embedded in clinical workflows that become routine even after their study periods are over. The RHAMI may uniquely be used to help healthcare systems rapidly identify such promising scalable solutions that address the gaps and priorities most of interest to local systems (and most aligned with their strategy and local values), rather than trying the mostly slower and costlier approach of uncritically experimenting with various AI precision nutrition applications.

## 5. Sovereign Swarm Agentic AI Social Networks for Sustainable Healthy Diets

AI-accelerated nutrition assessments tell you what to fix, and precision nutrition shows you how, but agentic AI may help you make these changes via healthy diets [39]. AI chatbots like ChatGPT can answer your questions, while AI agents can solve them, autonomously performing multi-step reasoning, carrying out goal-directed planning, adapting to changing conditions, and iteratively improving results with minimal human direction. Yet they are much more expensive to build and maintain, amid escalating geopolitical tensions and strained supply chains dividing and even balkanizing the global AI ecosystem (working to advance large-scale agentic AI through the prerequisite innovations technically and financially). So, there is a concurrent rise in platforms for sovereign agentic AI as a service with the US company NVIDIA, which is by far the leading provider of such enterprise AI factories or full-stack AI (the layered end-to-end architecture of chips, data centers, algorithms, workers, and governance as a resilient, self-reliant, and self-contained AI ecosystem). NVIDIA continues to power our AI era by providing countries and companies worldwide, including in LMICs such as India and those in the Middle East, this full-stack AI locally, especially for healthcare systems facing more stringent regulations than in other economic sectors. Instead, in sovereign AI, the data and models are constructed, and computations are carried out locally and thus more securely and affordably (compared to users having to build these stacks themselves as history’s most labor-, capital-, and expertise-intensive infrastructures ever)—including at the edge of the global digital ecosystem—rather than ferrying those assets across state borders to other countries’ cloud centers. Look at big hospitals, and you will often see power plants not far away, as the electric grid feeding them is part of the critical infrastructure enabling healthcare systems to operate (i.e., operating rooms cannot lose lighting in the middle of surgeries). It appears increasingly not far from now that hospitals will have to have neighboring AI factories as their critical digital infrastructure so they can process, analyze, and leverage their massive datasets and streams to provide care, including in nutrition. The decentralized counterpart to such sovereign AI may be swarm intelligence, which is rising in healthcare and is poised to break into nutrition in the near term [40]. Inspired by self-organized systems like ant colonies naturally, swarm agentic AI can collectively execute complex tasks, often secured through blockchain to keep data locally secure while sharing insights across the network. Stanford took this a step further, partnering with the private American software company Microsoft to create healthcare’s first multi-AI agent orchestrator in multi-center deployment in real-world clinical care [41]. With embedded HIPAA-compliant responsible governance, this AI compresses hours of complicated work per patient into minutes to integrate multimodal health and biologic data into actionable diagnosis and treatment recommendations as a digital tumor board. A traditional tumor board is a time- and resource-costly collaboration of multiple medical specialists to design a complex care plan—which is usually only performed for a small segment of patients with complicated or rare cancers served by large healthcare systems and mostly in HICs. Microsoft’s orchestrator, in contrast, coordinates diverse AI agents (each for the patient’s history, imaging, pathology, cancer staging, relevant clinical guidelines and clinical trial opportunities, latest research, and report creation, securely connecting the ambient EHR intelligence with real-time data from the global digital ecosystem for precision medicine). While nutrition appears to be moving there, it is just beginning with agents, including AI nutritionists and health coaches. A 2025 Dutch team demonstrated a globally scalable approach applicable for low-resource communities and LMICs with Meta’s freely available LLaMA3 to provide personalized, sustainable, evidence-based healthy meal plans [42]. By integrating authoritative dietary guidelines and the United Nation Sustainable Development Goals, the edge “virtual nutritionist” can make real-time recommendations on customized meals worldwide for users regardless of their health literacy using this local AI and locally sourced ingredients (keeping user data private and secure). Meanwhile, the Bill & Melinda Gates Foundation is funding Dimagi South Africa to create an AI coach for Malawi to empower LMIC front-line healthcare workers with LLM-generated personalized coaching—spanning early childhood development, skin-to-skin contact, financial management, and self-care—to holistically boost healthy eating for mothers and newborns [43].

Such interventions understand that improving healthy behaviors is best performed in the community, where individual changes are reinforced by others in one’s own social network (with the inverse being true also). The landmark network study from the *New England Journal of Medicine* had shown, through the Framingham Heart Study cohort across over 30 years and 12,067 people, that obesity spreads by up to 3 degrees of separation—independent of biology and geography—to the point that individuals’ chance of becoming obese increased by nearly 60% if their friends did first [44]. Leveraging such network dynamics for healthy eating, one of this review’s authors (D.J.M.) co-founded culinary medicine in the world’s first medical school-based teaching kitchen for underserved communities as an end-to-end, cost-efficient, and culturally sensitive public health nutrition intervention (augmented by novel AI agents for real-time continuous analysis and health improvement) [45]. The Bayesian adaptive randomized controlled trial within the largest longitudinal nutrition education study he created demonstrated that families in low-resource communities could lower their weekly food bills while tripling their odds of achieving healthy eating habits compared to the standard of care, while the medical students and physicians teaching those cooking classes had at least 5-fold increased odds of having healthy eating habits themselves after going through the course (which has since been implemented at over 55 healthcare systems, medical schools, and universities across the US). The leading AI platforms also understand the compounding effect of community. Meta’s edge AI-enabled smartglasses enable hands-free cooking with personalized meal plans, networked into its sister platform, Facebook, as the world’s largest social media network used by nearly half of the world’s population stretching across the Global South as well [46]. Its mature enterprise-wide AI already leverages AI agents for automated productivity, reinvesting its historic profits into historic capital expenditures explicitly seeking to be the first platform to turn its agents into AGI (artificial general intelligence capable of performing all human cognitive tasks *as good* as humans) and ultimately ‘personal’ superintelligence (doing so *better* than humans) to electrify the world’s largest digitally connected social network. Think of it as the smartest AI orchestrator of a swarm of edge AI on each digital device, streamlined in your day, supposedly empowering your healthy diets and lifestyles that are reinforced by like-minded friends and family in your community, and networked into families and communities worldwide collectively learning at unprecedented speed—amid significant critiques that AGI governance is even possible, or superintelligence can be controlled, let alone that it can fairly benefit populations outside the small number of massive technology companies rushing to build it, along with humanity’s future that supposedly will run on it.

In RHAMI analysis, Stanford’s Microsoft AI orchestrator provides a potential responsible AI-driven cost-saving democratization of one of healthcare’s most time- and resource-intensive services—the tumor board’s complex multi-specialty cancer care—in a way that can democratize and scale such services to lower-resource communities and LMICs, along with the less complex but more societally impactful services as boosting population-level healthy eating. Microsoft’s Responsible AI compliance by design embedded in this technology already has global scale by the company’s status as the world’s largest software provider, including in healthcare systems. Along with AI nutritionists and coaches especially promising for the Global South, there is a clear pathway for private AI platforms provisioning best-in-class sovereign swarm agentic AI amplified by social networks to collectively and thus sustainably advance healthy diets at the population and global scales. Yet the much higher profit margin for their use in specialized healthcare services like complex oncology means that strategic, deliberate, and collaborative focus is required by executives and front-line healthcare workers to selectively bring these technologies to bear to realize enterprise-wide AI throughout systems. No healthcare systems in HICs or LMICs feature even sustained organization-wide programs boosting healthy eating within essential public health nutrition as part of routine healthcare. So the most likely path to reverse this is for the continued rise in increasingly capable and versatile agentic AI, paired with smart edge devices like users’ smartphones whose adoption rate is growing the fastest in LMICs as rapid and decentralized swarm learning in ways that are useful, relevant, impactful, and still secure for users (with healthcare systems progressively integrating these largely private platform services into their existing clinical workflows as sovereign ambient intelligence).

## 6. Discussion

This narrative review attempts to provide a concise but novel, actionable, and global outline for the emergent roadmap helping to move nutrition research and practice from what may work to what stably scales in AI-accelerated public health nutrition driven by healthcare systems. Persistently urgent and unmet population health needs especially in LMICs demonstrate that it is not enough for AI research on nutrition to be about interesting use cases. Executives and clinicians need a rigorous full-spectrum analysis of how to scale the best ones throughout their local healthcare systems. To move from nutrition research experimenting with AI to transforming nutrition with AI through more mature translational research, this paper details and applies the novel AI meta-index of the RHAMI to generate a uniquely comprehensive and actionable healthcare tool to standardize the qualifying process in responsible enterprise-wide AI capabilities and functionalities for the continuous optimization of strategic planning and daily operations. Like building useful health AI, strategically implementing, growing, and maturing it within healthcare systems—where most health services, research activities, and societal funding occur—are increasingly complex, costly, and confusing processes for most systems. With shrinking financial and human capital resources, there is already a global and quickly growing practical need to rapidly identity, implement, integrate, innovate, and institutionalize AI-accelerated public health nutrition within healthcare systems that saves more lives (than the other interventions for which resources could be prioritized), saves more money (than it spends for patients, systems, and populations), and saves more of our common home (than the ecological resources required for less sustainable interventions). The RHAMI provides a potentially high-impact, practical, and scalable democratization, especially for lower-resource communities globally, to automatically synthesize the major domains of data inputs required for effective, efficient, and equitable solution outputs to build human-centered AI-accelerated healthcare systems, beginning with the public health nutrition essential for humanity’s health. Built collaboratively by front-line clinicians, public health practitioners, nutritionists, engineers, executives, and policymakers with decades of experience in direct service and system strategy, it is engineered to be used cooperatively in the increasingly globalized health ecosystem.

There are recent robust narrative reviews of AI’s role in nutrition assessments and healthy eating [34,47,48,49]. But their utility is limited for healthcare workers, leaders, and policymakers practically advancing the field for healthcare, public health, and global health. This gap between analysis and application arises from their descriptions of largely isolated nutrition AI use cases for nutrition detached from an empirical analysis within healthcare systems of (a) their readiness to integrate affordable nutrition AI into existing workflows and organizational structures, (b) their maturity in building sufficiently sophisticated and scaled nutrition AI throughout their enterprises, and (c) their governance of nutrition AI to ensure its ethical, regulatory, legal, and professional compliance for sustainable net benefit for the organizations and the health communities they serve. They also lack the specific focus on underserved communities and LMICs who stand to most benefit from such nutrition AI and who face the greatest disparities in receiving them. Additionally, the AI indexes cited in the methodology attempting to make AI useful not just useable for such urgent health challenges are influential but also limited in their practical benefit for healthcare systems. They focus on AI readiness, maturity, or responsibility (but fail to meaningfully unite them for end-to-end decision-making). And they provide high-level guidance but not operationally significant insights in system administration or clinical care to actualize it. Instead, this paper uniquely provides the first multi-dimensional, multi-disciplinary, and multi-sector narrative review through a novel end-to-end automated health AI index for AI-enabled nutrition assessment and healthy eating by an international team of front-line clinicians, ethicists, engineers, and leaders. Such an integration is necessary (and increasingly inevitable given worsening resource constraints on the current trajectory pressuring more seamless analysis and leadership). Even in the US with the most mature national AI ecosystem and richest national health sector, only 18% of healthcare systems have mature AI, less than half report sufficient resources just to begin the implementation of AI, most are making or moving toward AI investments principally to reduce costs, and over 80% report insufficient capabilities to identify and integrate AI within their system [50]. Nearly 80% report that they depend on their current vendors (or their AI partners) to implement AI given their lack of endogenous capabilities to do so. Without an end-to-end, empirically verifiable approach embodied by the RHAMI, there is no clear pathway to scaling responsible and mature AI for nutrition let alone health globally where it is most needed.

Compared to prior studies in this field, the RHAMI’s results on national AI ecosystems and AI-enabled healthcare systems are collectively comparable to other indexes and analyses focused on more isolated aspects of their AI [12,51,52]. But it goes further to provide this end-to-end approach for responsible AI readiness and maturity evaluation, including at the global level applied to nutrition AI (with more granular and real-time insights for local healthcare systems’ clinicians and executives according to their particular needs). The analysis does support how Mayo Clinic appears to likely remain the frontrunner in mature health AI including on nutrition. Cleveland Clinic is positioning itself to further compete, especially if it can deliver on its strategic partnership announced in May 2025 with the Emirati’s G42 and Oracle Health) for the first nation-scale and most advanced AI-accelerated healthcare platform [53]. This trajectory analysis technically derives from Oracle’s capabilities as the world’s largest database management system (operating the first cloud-native EHR with embedded voice-first agentic AI, second only to Epic in global market share of EHRs), along with G42’s status as a leading sovereign AI infrastructure company within the Global South (also operating 480 clinics in 26 countries and the world’s largest genomic program for nation-scale implementation of precision medicine) [54].

There are several key limitations to this narrative review. Given the broad scope of the topic and limited space in this paper, some degree of depth had to be sacrificed to allow for the sufficient scope of the methodology and results on the leading nutrition AI use cases manifesting the larger structural trends (yet to improve such analyses, and the AI index methodology doing so, much greater detail is required to allow for sufficient peer and public scrutiny of the index to improve it, which will be implemented in the next paper). More granular country- and system-specific analyses are also required, which the next phase of this study pipeline will conduct with an open-source API to address this. Also, this analysis lacked a more detailed analysis of China, for instance, which has a booming national AI ecosystem second only to the US but limited investments and expenditures in its national health system (half the global per capita average as a percentage of their GDP) and an even lower scaling of responsible AI governance throughout it—though China still has significant innovations as noted above, along with a world leading position in AI infrastructure [55,56]. Notwithstanding such limitations, this narrative review still concisely provides a unique full-spectrum overview of the state of the art in scaling AI-accelerated innovations by healthcare systems for nutrition assessment and healthy eating (with particular focus on low-resource communities and LMICs) using a novel end-to-end AI index to automate future optimization by front-line clinicians, leaders, and policymakers. The smarter our AI methods and interventions become in health nutrition, the more they appear to facilitate us getting back together at the dinner table to recover what health and community mean.

## Figures and Tables

**Figure 1 nutrients-18-00038-f001:**
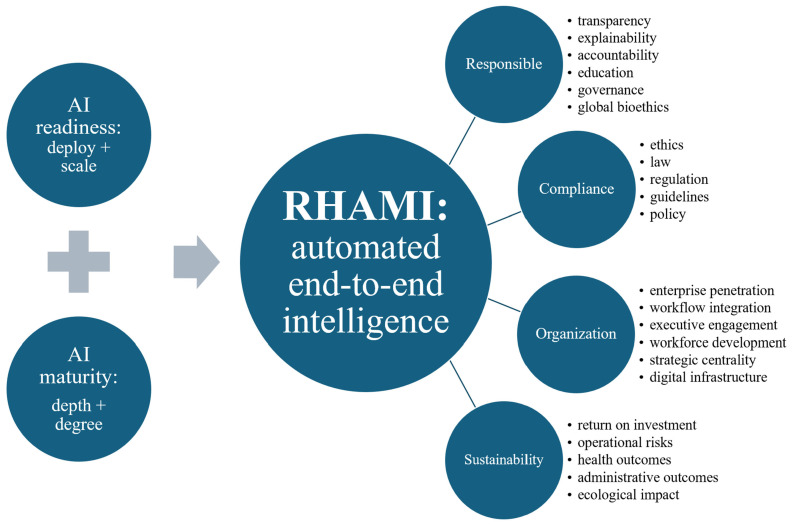
RHAMI embedded in health systems for comprehensive, continuous, and automated optimization of AI-accelerated strategy and operations.

## Data Availability

No new data were created or analyzed in this study. Data sharing is not applicable to this article.

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
