# Peer review of "The Responsible Health AI Readiness and Maturity Index (RHAMI): Applications for a Global Narrative Review of Leading AI Use Cases in Public Health Nutrition"

_nutrients, 2025, doi:10.3390/nu18010038_

Round 1
Reviewer 1 Report
Comments and Suggestions for Authors
The manuscript title is excessively long. ‘Poor diet is the leading risk factor for death worldwide associated with over 10 million premature deaths annually [1]’ – make sure you support such claims by the most recent data, if no specific year is mentioned. ‘Artificial intelligence or AI’ - Artificial intelligence (AI). ‘The World Health Organization (WHO) in its Nature 2025 publication’ – specific reference number is needed. Also avoid mentioning the journal title for in-text style consistency. There are several such instances. ‘This is especially true in nutrition which may be among the most effective fulcrums of public health advanced by healthcare systems’ – need substantiation. ‘An AI unassisted comprehensive literature review was performed from 2005 to 2025 using EBSCO, Web of Science, Scopus, Google Scholar, PubMed’ - Google Scholar does not filter content in terms of quality. What quality tools (e.g., CADIMA, JBI SUMARI, Litstream, METAGEAR package for R, Nested Knowledge, SWIFT-Active Screener, etc.) did you use in article selection? ‘This culturally-sensitive ambient intelligence enables more precise and personalized nutrition assessments and thus recommendations for healthy eating based on tracked individual food preferences, portion sizes, and meal timing’ – is this your conclusion/analysis or [23]’s? There are several such instances. E.g., ‘Nutri rapidly assesses and personalizes recommendations from the latest guidelines and research that clinicians can then modify and apply for achievable diet modifications, while complementing rather than disrupting the other core aspects of clinical visits’, ‘It serves as the top nutrition app on the digital edge with over 250 million users in 120 countries, generating personalized nutrition plans from over 5 million foods based on this free tracker (for calories, macronutrients, micronutrients enabled with computer vision, along with the fee-based subscription including barcode scanning and voice logging also)’, etc. ‘A 2024 Chinese randomized controlled trial across 11 hospitals and 5,763 patients showed that an AI-based rapid nutritional diagnostic system embedded within routine clinical care significantly improved malnutrition diagnosis and reversal, at an AI intervention cost that is 73% cheaper than China’s per capita cost of malnutrition’s health effects’ – missing source. ‘This trend is fueled by the historic successes in particular of precision medicine for cancer care to translate granular knowledge of individuals’ genes, lifestyles, and environments into targeted treatments with maximum clinical benefit, minimum risk, and optimized overall affordability (uniting genomics, epigenomics, proteomics, metabolomics, and other -omics biological fields to facilitate healthier states faster, thus reducing often unaffordable longitudinal clinical and financial costs of disease)’ – need substantiation. ‘(which also houses the Stanford Institute of Human-Centered AI as a world leader in responsible AI governance’ – remove, irrelevant. ‘There are a number of recent robust narrative reviews of AI’s role in nutrition assessments and healthy eating [34, 47-49]’, ‘Compared to prior studies in this field, RHAMI’s results on national AI ecosystems and AI-enabled healthcare systems are collectively comparable to other indexes and analyses focused on more isolated aspects of their AI [12, 51-52]’ – develop and clarify the specific contribution of each cited source. General synopsis of evidence as regards focus topics and descriptive outcomes is needed. In the Discussion section you should focus on comparisons with other research outcomes, as recent and relevant as possible. About half of the cited sources are not from peer reviewed journals.
Author Response
Reviewer comments in non-bold type, and responses in bold type:
- The manuscript title is excessively long. We deeply appreciate the reviewer’s generous contribution of the reviewer’s time and expertise for the detailed and helpful critique. We sought to provide point-by-point responses and revisions (which we hope sufficiently strengthens the paper to justify publication in this esteemed journal). The title was thus revised, along with the below revisions also made.
- ‘Poor diet is the leading risk factor for death worldwide associated with over 10 million premature deaths annually [1]’ – make sure you support such claims by the most recent data, if no specific year is mentioned. Confirmed and cited accordingly (we are awaiting the updated GBD analysis or comparable since the reference publication, but to our knowledge, no widely recognized and trusted analysis has revised, refined, or refuted the above result).
- ‘Artificial intelligence or AI’ - Artificial intelligence (AI).
- ‘The World Health Organization (WHO) in its Nature 2025 publication’ – specific reference number is needed.
- Also avoid mentioning the journal title for in-text style consistency. There are several such instances.
- ‘This is especially true in nutrition which may be among the most effective fulcrums of public health advanced by healthcare systems’ – need substantiation.
- ‘An AI unassisted comprehensive literature review was performed from 2005 to 2025 using EBSCO, Web of Science, Scopus, Google Scholar, PubMed’ - Google Scholar does not filter content in terms of quality. What quality tools (e.g., CADIMA, JBI SUMARI, Litstream, METAGEAR package for R, Nested Knowledge, SWIFT-Active Screener, etc.) did you use in article selection? ‘ Updated (in accordance with the existing literature on the topic, as discussed by Chigbu et al., 2023 in Nutrient’s MDPI sister journal, more extensive use of such quality screening tools including software analysis tools was not included given this is a narrative not systematic review).
- This culturally-sensitive ambient intelligence enables more precise and personalized nutrition assessments and thus recommendations for healthy eating based on tracked individual food preferences, portion sizes, and meal timing’ – is this your conclusion/analysis or [23]’s? Informed by the conclusions of [23], the above observation is our consensus observation crafted in accordance with accepted practice for narrative reviews, as detailed emblematically in Chigbu et al., 2023 above to provide readers hopefully a sufficiently broad, flexible, actionable, and clinically useful overview of the existing literature on this topic.
- ‘A 2024 Chinese randomized controlled trial across 11 hospitals and 5,763 patients showed that an AI-based rapid nutritional diagnostic system embedded within routine clinical care significantly improved malnutrition diagnosis and reversal, at an AI intervention cost that is 73% cheaper than China’s per capita cost of malnutrition’s health effects’ – missing source.
- ‘This trend is fueled by the historic successes in particular of precision medicine for cancer care to translate granular knowledge of individuals’ genes, lifestyles, and environments into targeted treatments with maximum clinical benefit, minimum risk, and optimized overall affordability (uniting genomics, epigenomics, proteomics, metabolomics, and other -omics biological fields to facilitate healthier states faster, thus reducing often unaffordable longitudinal clinical and financial costs of disease)’ – need substantiation.
- ‘(which also houses the Stanford Institute of Human-Centered AI as a world leader in responsible AI governance’ – remove, irrelevant.
- ‘There are a number of recent robust narrative reviews of AI’s role in nutrition assessments and healthy eating [34, 47-49]’, ‘Compared to prior studies in this field, RHAMI’s results on national AI ecosystems and AI-enabled healthcare systems are collectively comparable to other indexes and analyses focused on more isolated aspects of their AI [12, 51-52]’ – develop and clarify the specific contribution of each cited source. In keeping with accepted practice for narrative reviews (as detailed in Chigbu et al., 2023), we attempted to develop the statement regarding the 34 and 47-49 reviewers in the subsequent lines regarding their shared traits (as individually detailing their limitations for the topic of our narrative review and its intended audiences and applications would excessively expand the scope and provide unnecessary detail amid word count limitations). A similar rationale and approach to substantiating the statement in regards to references 12 and 51-52 also applies, including with the subsequent lines attempting to detail the unique contributions this review applies in contrast to the above cited ones.
- General synopsis of evidence as regards focus topics and descriptive outcomes is needed. We definitely agree, and thus sought to provide this overview for the focus topics (ambient intelligent nutrition assessments, precision nutrition platforms, and swarm agentic AI-supported healthy diets) and descriptive outcomes (detailed each for the included studies as applicable) in keeping with accepted practice for narrative reviews (as detailed in Chigbu et al., 2023).
- In the Discussion section you should focus on comparisons with other research outcomes, as recent and relevant as possible. We similarly agree and appreciate the reviewer’s helpful point on this. We attempted to do so per the above response. Given the accepted practice for narrative reviewers in contrast to systematic reviewers or meta-analyses (which our team has published elsewhere and previously on other health topics), we sought to provide a broad and flexible overview of the focus topics that spans a wide range of research and applied outcomes as relevant for the target audiences.
- About half of the cited sources are not from peer reviewed journals. In keeping with accepted practice for narrative reviews (as detailed in Chigbu et al., 2023), we sought to focus on publications from peer-reviewed journals to improve the quality of the latest reported evidence on the focus topics. But given the fast moving developments in AI-assisted nutrition and the lower peer reviewed publications detailing developments especially in low- and middle-income countries, we attempted to fill such gaps through citation of widely referenced sources (like the United Nations and the International Monetary Fund) in addition to peer-reviewed academic book publications including those by Monlezun (as published by Elsevier as the world’s largest academic publisher).
Reviewer 2 Report
Comments and Suggestions for Authors
-
While I find the motivation of the study presented in the introduction convincing, the research problem itself is not clearly articulated. In addition, the proposed research gap, research questions, and the intended academic and practical contributions are not sufficiently or systematically presented. These elements should be more explicitly and coherently structured.
-
The research methodology lacks sufficient clarity, making it difficult to fully assess or reproduce the study. The authors are encouraged to provide a more detailed and transparent description of the research design, data sources, analytical procedures, and overall methodological framework.
-
In particular, Figure 1 appears overly abstract at its current stage. The conceptual components and their interrelationships require clear theoretical and/or empirical justification to strengthen the validity and explanatory power of the proposed model.
-
Overall, the study’s methodology seems to rely on insufficient theoretical and statistical grounding, which limits the robustness of the proposed framework and the credibility of the findings. Strengthening the linkage to relevant theories, prior literature, and sound statistical rationale would significantly enhance the rigor of this work.
Author Response
Reviewer comments in non-bold type, and responses in bold type:
- While I find the motivation of the study presented in the introduction convincing, the research problem itself is not clearly articulated. In addition, the proposed research gap, research questions, and the intended academic and practical contributions are not sufficiently or systematically presented. These elements should be more explicitly and coherently structured. Thank you for the kind consideration, along with your appreciated expertise and helpful recommendations here. In keeping with accepted practice for narrative reviews (as detailed in Chigbu et al., 2023 in Nutrient’s MDPI sister journal), we sought to build on our last narrative review published by Nutrients last year (https://www.mdpi.com/2072-6643/16/16/2601) to provide an updated and novel narrative review that would hopefully warrant publication in this esteemed journal. We therefore updated in this narrative review the focus topics in this fast-moving field, along with the methodology for this narrative review using the RHAMI index to augment the traditional and accepted practices of academic narrative reviews. To address your helpful above point, we sought to better clarify these points in the introduction, in addition to revising the section headers to make it clearer that this is meant to be a narrative review, rather than a systematic review or meta-analysis of these topics, or an original article with extensive theoretical and empirical detail of the index’s design, deployment, validation, and comparison to prior indexes.
- The research methodology lacks sufficient clarity, making it difficult to fully assess or reproduce the study. The authors are encouraged to provide a more detailed and transparent description of the research design, data sources, analytical procedures, and overall methodological framework. We definitely agree with these astute points by the reviewer. Per the above response, and in accordance with accepted practices for academic narrative reviewers, we attempted to provide the necessary relevant details for the index as it relates to its application to the narrative review. To preserve the review’s focus on leading AI uses cases in public health nutrition, along with the special issue’s focus on the same (for which the lead author, Dominique Monlezun, was invited by Nutrients to serve as the Guest Editor), we were concerned with further expansion of the methodological description of the index (already at 3 paragraphs and greater than most narrative review’s methodological descriptions). We definitely agree though that further and wider use, testing, critique, reproducibility, and improvement on the index including by other research teams require significantly more methodological detail. That is why we are drafting a separate research study to provide such at the request of a different peer-reviewed journal currently, but it is outside the scope of Nutrients and this special issue, and thus we did not include or reference it in this narrative review.
- In particular, Figure 1 appears overly abstract at its current stage. The conceptual components and their interrelationships require clear theoretical and/or empirical justification to strengthen the validity and explanatory power of the proposed model. We definitely agree. If the reviewer would kindly permit us to reference the above response that we hope sufficiently clarifies our rationale.
- Overall, the study’s methodology seems to rely on insufficient theoretical and statistical grounding, which limits the robustness of the proposed framework and the credibility of the findings. Strengthening the linkage to relevant theories, prior literature, and sound statistical rationale would significantly enhance the rigor of this work. We agree with the reviewer’s helpful point. Per the above responses, we attempted to provide references for the prior indexes that informed the development of this one, while limiting further detail to only what we hope is sufficient to apply to this narrative review and the special issue’s focused scope (as the 3 paragraphs on the index are already notably longer than most narrative reviews, and so we hoped to avoid unnecessarily distracting readers away from the review’s focus).
Round 2
Reviewer 1 Report
Comments and Suggestions for Authors
Acceptable.
Reviewer 2 Report
Comments and Suggestions for Authors
The revision is fine. I recommend that the paper is now acceptable.